# Androgen Effects on Amyloid Precursor Protein Processing Pathways in Cancer: A Systematic Review

**DOI:** 10.3390/cimb47121041

**Published:** 2025-12-12

**Authors:** Mai Alhadrami, Gideon Stone, Rachel M. Barker, Jennifer C. Palmer, Patrick G. Kehoe, Claire M. Perks

**Affiliations:** 1Cancer Endocrinology Group, Learning & Research Building, Southmead Hospital, Translational Health Sciences, Bristol Medical School, Bristol BS10 5NB, UK; oc21138@bristol.ac.uk (M.A.); claire.m.perks@bristol.ac.uk (C.M.P.); 2Cerebrovascular and Dementia Research Group, Learning & Research Building, Southmead Hospital, Translational Health Sciences, Bristol Medical School, Bristol BS10 5NB, UK; gideon.stone@bristol.ac.uk (G.S.); patrick.kehoe@bristol.ac.uk (P.G.K.); 3NIHR Bristol Evidence Synthesis Group, Population Health Sciences, Bristol Medical School, Bristol BS10 5NB, UK; jen.palmer@bristol.ac.uk

**Keywords:** androgens, amyloid precursor protein and prostate cancer

## Abstract

Androgens have been shown to be linked to cancer progression, particularly in hormone-dependent cancers such as prostate and breast cancer, but also other cancers. Amyloid precursor protein (APP), which has primarily been studied in Alzheimer’s disease, is gaining recognition for its role in tumor growth and survival. While APP overexpression and androgen receptor (AR) signaling are each associated with cancer progression, the connection between androgens and APP processing in cancer has not been thoroughly investigated. This systematic review was conducted through a comprehensive search of PubMed, Scopus, Web of Science, and EMBASE between 2000 to 2024 for studies examining the effects of androgens on APP and its cleavage enzymes in cancer. Five experimental studies met the inclusion criteria, covering prostate and breast cancer models. Data were extracted and synthesized narratively due to heterogeneity in methods and outcomes. Three studies reported that dihydrotestosterone (DHT) or AR agonists increased the expression and nuclear translocation of ADAM10, a key α-secretase enzyme in the non-amyloidogenic APP processing pathway. Two studies identified APP as an androgen-responsive gene, showing that androgens upregulated APP expression in prostate and breast cancer cells and promoted the proliferation of cancer cells. Inhibition or knockdown of APP and ADAM10 reduced proliferation, supporting their roles in tumor progression. Androgen signaling modulates APP processing in cancer, particularly through the non-amyloidogenic pathway; however, significant knowledge gaps remain. Further studies are needed to explore the interaction between androgens and APP processing in other cancer types, as well as to elucidate downstream signaling pathways regulated at the gene expression level.

## 1. Introduction

Cancer is the second leading cause of death worldwide. In 2022, there were ~20 million new cases and ~9.7 million cancer deaths globally [1]. Among these, female breast cancer is known as the most commonly diagnosed cancer related to women, which accounted for 2.3 million new cases and ~666,000 deaths. Prostate cancer, which is recognised as one of the most common cancers in men, was responsible for 1.5 million new cases and 397.000 deaths worldwide [1]. Additionally, in 2021, prostate cancer contributed approximately 8.14 million disability-adjusted life years (DALYs) globally [2], while breast cancer accounted for around 20.64 million DALYs [3]. These statistics highlight the substantial global burden of these malignancies and underscore the importance of identifying potential targeted therapies that may influence disease progression.

Androgens, a class of steroid hormones, have been shown to contribute to cancer progression, especially in hormone-dependent prostate and breast cancers [4,5]. Androgens are particularly important for the growth and regulation of the human prostate. Testosterone is considered the primary androgen because it is the most abundant and biologically significant androgen in circulation, serving as the precursor for more potent metabolites such as 5α-dihydrotestosterone (DHT), which is formed by the action of the enzyme 5α-reductase [6]. DHT binds with high affinity to the androgen receptor (AR), which subsequently enables the AR to translocate into the nucleus [7]. This nuclear translocation activates the androgen signalling pathway, driving the transcription of target genes that promote cell proliferation and survival (Figure 1) [8], including cyclin D1 and E, to promote cell cycle progression [9], Bcl-2 and Bcl-xL, which inhibit apoptosis [10] and the PI3K/AKT and RAS/MAPK pro-survival signalling pathways [11]. Androgens modulate the growth of several cancer types beyond prostate cancer, including breast [12], bladder [13,14], and lung cancer [15]. In breast cancer, AR signaling exerts context-dependent effects: it can inhibit proliferation in estrogen-receptor-positive (ER-positive) tumors by antagonizing ER activity, but in ER-negative subtypes such as the luminal androgen receptor (LAR) triple-negative breast cancers, AR activation promotes growth through PI3K/AKT and MAPK pathways [12]. While in bladder cancer, AR drives cell cycle progression and invasion via upregulation of proliferation-related genes and crosstalk with EGFR signaling [13,14]. In lung cancer, AR influences proliferation and therapy sensitivity through interactions with EGFR and MAPK/ERK pathways, with effects varying by tumor subtype and AR splice variant expression [15].

Amyloid precursor protein (APP), more renowned for its involvement in Alzheimer’s disease (AD), is the precursor of the amyloid-beta (Aβ) peptide, which is believed to be heavily implicated and a cause of neurotoxicity in AD pathogenesis [16]. APP is known to be processed through two pathways with respect to Aβ. The non-amyloidogenic pathway, mediated by α- and γ-secretases, precludes the formation of Aβ, and the amyloidogenic pathway, involving sequential cleavage by β- and γ-secretases to produce Aβ, is upregulated in AD (Figure 2) [17]. Although APP is predominantly studied for its role in the brain—particularly in AD—evidence indicates that it exerts broader biological functions. These include influencing cell proliferation, differentiation, and cell adhesion [18]. Consistent with these functions, APP has been reported to be overexpressed in various cancers, including glioblastoma, breast, prostate, pancreatic, lung, and colon cancers [19]. This overexpression has been linked to critical cancer-related processes such as tumour growth, metastasis, and disease progression [19]. Importantly, the non-amyloidogenic pathway may be particularly relevant in cancer, as its N-terminal product, sAPPα, has been shown to promote cell proliferation and survival in previous studies [20,21]. Unlike sAPPα, the role of Aβ in cancer promotion or progression remains uncertain. The current studies are limited and somewhat conflicting, with some indicating that Aβ oligomers can cause cancer cell death [22]. However, other research shows Aβ accumulation in the tumor microenvironment, including near blood vessels [23], or increased levels in specific cancers such as hepatocarcinoma [24]. These conflicting results emphasize the need for more research to determine whether Aβ promotes or opposes cancer progression.

While the role of androgens in cancer, particularly prostate cancer, has been extensively studied, and APP has also been implicated in several androgen-related cancers, the potential interaction between androgens and APP processing pathways is less clear. The aim of this systematic review is to summarize the current scientific evidence on the effects of androgens on APP processing in cancers, with the hope of identifying new potential therapeutic targets and highlighting critical knowledge gaps where further research is needed.

## 2. Materials and Methods

A protocol for this review was submitted to PROSPERO and registered under the number CRD42023433147. https://www.crd.york.ac.uk/PROSPERO/view/CRD42023433147 (accessed on 6 June 2023). The review process was conducted and reported following the PRISMA guidelines to ensure transparency and methodological rigor to minimize bias and provide a comprehensive and reliable summary [25]. The completed PRISMA checklist is shown in Appendix A.

### 2.1. Search Strategy

We conducted a systematic search across four databases: PubMed, Web of Science, Scopus, and Embase, to find studies investigating the role of androgens in APP processing pathways in cancer published between (01/01) 2000 and (12/31) 2024. The PubMed search utilized the following query: ((androgens) OR (testosterone)) AND ((amyloid precursor protein) OR (BACE1) OR (BACE2) OR (ADAM10) OR (ADAM17) OR (TACE) OR (Metalloproteases 10) OR (Metalloproteases 17)) AND ((cancer) OR (tumor) OR (tumour) OR (neoplasia)). For the other databases, we applied a similar query, structured as follows: ((androgens OR testosterone) AND (amyloid precursor protein OR BACE1 OR BACE2 OR ADAM10 OR ADAM17 OR TACE OR Metalloproteases 10 OR Metalloproteases 17) AND (cancer OR tumor OR tumour OR neoplasia)).

### 2.2. Search Criteria

Inclusion Criteria: (1) Studies investigating the effects of androgens on APP and its cleavage enzymes (α-secretase and β-secretase), including in vitro, in vivo, and clinical research. (2) All cancer types. (3) Data on androgen effects, such as changes in mRNA expression or protein levels of molecules associated with APP processing pathways. (4) Peer-reviewed journal articles published in English.

Exclusion Criteria: (1) Studies examining the effects of androgens specifically on Alzheimer’s disease (AD) and not cancer. (2) Studies examining the effects of androgens on anything other than the APP processing pathways. (3) Studies that are not investigating androgens. (4) Non-peer-reviewed articles, conference abstracts.

### 2.3. Study Selection and Data Extraction

For study selection, two independent reviewers first evaluated titles and abstracts using RAYYAN https://www.rayyan.ai/ (accessed on 29 October 2025), with conflicts resolved through discussion or by consulting a third reviewer. Full texts of articles included at this stage were retrieved and assessed for inclusion in the same way. Key information was extracted by one reviewer and checked by a second about the publication, type of cancer studied, methods employed, and the relevant results, such as reported levels of APP and/or α/β-secretase enzymes.

### 2.4. Assessment of Risk of Bias

To assess the quality and potential risk of bias in the included studies, we developed a predefined set of criteria focusing on key methodological aspects: use of in vivo models, human tissue samples, clinical data integration, mRNA and protein expression measurement, methodological complexity, and mechanistic insights. We scored each study across these domains, with a total score categorizing studies as high, moderate, or low quality. Higher scores indicate robust methodologies with a lower risk of bias, whereas lower scores reflect potential biases due to limited experimental design, lack of mechanistic insights, or reliance on fewer methodologies. These criteria were created specifically for this review and have not been published elsewhere.

### 2.5. Data Synthesis

Due to the heterogeneity in study designs, experimental models, and outcome measures, meta-analyses were not feasible. Instead, we conducted a narrative synthesis to integrate findings from the five included studies. We synthesized results by identifying common themes across studies, including trends in molecular expression changes, mechanistic insights, and methodological approaches. Data were grouped based on key variables such as the use of human tissue, in vivo models, and mechanistic investigations. We considered the quality assessment scores to evaluate the reliability of reported findings.

## 3. Results

### 3.1. Study Selection

The study selection process is summarized in Figure 3. We identified 693 unique studies through our searches, excluded 677 studies at title and abstract screening, and reviewed the full texts of 16 studies. Five articles, the characteristics of which are summarized in Table 1, met our eligibility criteria, and we included them in our synthesis.

### 3.2. The Risk of Bias

As indicated by the quality assessment scores, which varied across the included studies (Table 2). Three studies (Arima et al., 2007, Takayama et al., 2009, and Takagi et al., 2013) scored highly (≥23/30), suggesting strong methodological rigor and a lower risk of bias [28,29,30]. Conversely, McCulloch et al., 2000 [26] had a low score (6/30), indicating a higher risk of bias due to the absence of in vivo models, clinical data, and mechanistic analysis. The remaining study had moderate scores (15–22/30) and some concerns, primarily due to limitations in mechanistic insights or methodological depth. These variations in study quality may introduce heterogeneity in the findings.

Of the five studies included, two were experimental in vitro studies using cancer cell lines, while three studies used human tissue samples (prostate and breast). For tissue-based studies, sex can be inferred from the tissue type (male for prostate, female for breast). However, information on race and age of the donor was not reported in the original studies. Therefore, potential biases related to patient demographics were limited.

### 3.3. Narrative Synthesis

Of the five included studies, three [26,27,28] published between 2000 and 2007 focused on the impact of DHT on ADAM10, a key α-secretase enzyme involved in the non-amyloidogenic pathway of APP processing. The remaining two studies [29,30], published in 2009 and 2013, investigated the effects of androgens on APP expression. Four studies were investigating prostate cancer [26,27,28,29], and one study was investigating breast cancer [30]. We found no studies investigating the role of androgens on APP processing in any other cancer types.

Across the five studies, common findings emerged regarding androgens, APP, and α-secretase enzymes, such as ADAM10. Three high-quality studies (Takayama et al., 2009, Takagi et al., 2013, Arima et al., 2007) consistently demonstrated that androgens such as DHT and R1881 promoted cancer growth, that was accompanied by an increase in the expression of APP and enhanced nuclear translocation of ADAM10, whereas the lower-quality studies (McCulloch et al., 2000 and 2004) showed variable results, potentially due to methodological differences [26,27,28,29,30]. Results from studies relying solely on in vitro models are difficult to interpret due to having limited translational relevance. Studies incorporating human tissue samples and clinical data provided stronger evidence. They indicated that high APP protein levels correlate with an aggressive cancer stage, and they observed strong nuclear staining for ADAM10 in prostate cancer tissues compared to benign prostate hyperplasia.

#### 3.3.1. Amyloid Precursor Protein (APP)

Two studies [29,30] explored the impact of androgens on *APP* expression in prostate and breast cancers. Takayama et al. [29] were the first to identify *APP* as an androgen-regulated gene, discovering an AR-binding site within its transcriptional region using the ChIP-chip technique. This study examined the effect of R1881 on *APP* expression and abundance in LNCaP cells. Treating with R1881 (10 nM/L) for 72 h significantly upregulated *APP* mRNA levels 2.5-fold at 24 h and 2.8-fold at 48 h. Moreover, an increase in APP abundance was observed in the presence of R1881 at all time points (24, 48, and 72 h). Additionally, the study explored the role of APP in cancer progression. Overexpression of *APP* significantly promoted the growth of LNCaP cells over 4 days (*p* < 0.05). Similarly, the addition of a soluble APP peptide (10, 25 nM/L) led to a notable increase in cell proliferation at 24 h (*p* < 0.05) for both concentrations and at 48 h (*p* < 0.01 and *p* < 0.001), respectively. Conversely, silencing *APP* with two different siRNAs (siAPP-A and siRNA-B) for 48 h at 200 nM/L inhibited the proliferation of androgen-dependent LNCaP cells significantly (*p* < 0.05). Furthermore, male mice implanted with LNCaP tumour cells exhibited significantly reduced tumour volume following siAPP-B silencing (*p* < 0.05).

The second study [30], conducted by Takagi et al., identified cytoplasmic localization of APP in breast carcinoma tissues and demonstrated a positive correlation between APP and AR levels. ER-positive breast cancer cells, MCF7s, treated with 10 nM DHT for 72 h, significantly increased *APP* mRNA levels (*p* < 0.01). In comparison, treating with an AR antagonist, hydroxyflutamide (10µM) to block the AR, decreased *APP* mRNA levels by 0.79-fold. Silencing *APP* for 1, 2, and 4 days in both MCF7 and MDA-MB-231 breast cancer cells led to a significant decrease in cell proliferation. In MCF7 cells, the decrease was observed at day 2 and day 4 (*p* < 0.001), while in MDA-MB-231 cells, the reduction occurred at day 1, day 2, and day 4 (*p* < 0.001). However, silencing *APP* did not induce apoptosis in these cell lines.

#### 3.3.2. Zinc-Metalloproteases (ADAM10)

Three studies focused on the AR-positive prostate cancer cell line LNCaP [20,21,22,26,27,28]. DHT, the active form of androgens, was utilized at concentrations ranging from 0.1 to 100 nM/L. The findings suggest that DHT modulated both the mRNA expression levels and nuclear protein translocation of ADAM10.

In the first study [26], McCulloch et al. demonstrated that treating LNCaP cells with DHT (0.1, 1, and 10 nM/L) for 96 h led to a moderate yet significant dose-dependent increase in cell proliferation, ranging from 7% to 23%. Furthermore, they examined the mRNA expression of three ADAM family members (*ADAM9*, *ADAM10*, and *ADAM17*) following 48 h of DHT treatment. They observed that *ADAM10* mRNA expression increased 2.5-fold at 0.1 nM, reached a significant 5-fold increase at 1 nM (*p* < 0.005), and then declined at 10 nM, though it remained 2-fold above the control levels. In contrast, *ADAM17* mRNA levels decreased at all concentrations, showing reductions of 1.6-, 1.7-, and 1.5-fold (*p* < 0.05), respectively. Meanwhile, *ADAM9* mRNA expression increased at 1 nM and 10 nM DHT by 1.5- and 1.7-fold (*p* < 0.05, *p* < 0.005), respectively.

The second study [27] by McCulloch et al. explored the effects of DHT (1 and 10 nM/L), insulin-like growth factor-I (IGF-I) (50 ng/mL), and epidermal growth factor (EGF) (50 ng/mL) on the proliferation of LNCaP cells. They observed that 96 h treatment with DHT alone increased the proliferation by 19%, with an additional 24% increase when DHT was combined with IGF1. Additionally, EGF (5, 10, 50 ng/mL) treatment for 96 h enhanced the growth of LNCaP cells significantly by 4-fold at the highest dose. The study also examined both mRNA expression and protein levels of *ADAM10* and investigated the localization of ADAM10 in benign and cancerous prostate tissue samples. Treatment with either IGF1 (50 ng/mL) or DHT (10 nM) alone reduced *ADAM10* mRNA levels significantly (*p* < 0.05 and *p* < 0.01), and in combination (i.e., IGF1 and DHT) significantly increased the mRNA level by 1.8-fold (*p* < 0.01). Furthermore, the immature (100 kDa) and mature (60 kDa) forms of ADAM10 protein were significantly upregulated in response to DHT (10 nM) combined with IGF-I (10 or 50 ng/mL), showing an increase of 1.8-fold and 4-fold (*p* < 0.05), respectively. Interestingly, a significant increase was observed in *ADAM10* mRNA, and protein abundance (*p* < 0.01) following dosing with EGF (50 ng/mL) alone.

Lastly, Arima et al. [28] were the first to investigate the subcellular localization of ADAM10 in response to DHT treatment in LNCaP cells. Their findings revealed that incubating cells for 24 h with DHT (0.1, 1, 10, 100 nM) promoted the translocation of ADAM10 from the cytoplasm to the nucleus. Specifically, the abundance of both the pro-form and active forms of ADAM10 decreased significantly at the two higher concentrations (10 and 100 nM) in the cytoplasmic fractions. In contrast, a notable upregulation of both ADAM10 forms (*p* < 0.05) was observed in the nuclear fraction in response to the same DHT concentrations (10 and 100 nM). Additionally, immunocytochemistry was performed to confirm this translocation, showing exclusive nuclear localization of ADAM10 in response to DHT. Interestingly, using immunoprecipitation, they were the first to report an association between the androgen receptor (AR protein) and ADAM10 in the nuclear fraction of LNCaP cells, and co-immunoprecipitated levels of AR were increased in a trend towards significance in a dose-dependent manner following DHT treatment (*p* = 0.09).

Moreover, benign prostate hypertrophy (BPH) (*n* = 20) and prostate cancer (PC) (*n* = 64) tissue samples were analysed to study the correlation of ADAM10 localization with clinical parameters such as Gleason score and preoperative prostate-specific-antigen (PSA) levels. They reported that ADAM10 was strongly stained along the cell membrane in BPH tissues compared to PC. Conversely, significantly strong staining (*p* < 0.0001) was observed in the nucleus of PC tissues. Furthermore, high-grade PC samples exhibiting strong nuclear staining significantly correlated with higher Gleason scores and PSA levels (*p* < 0.001 and *p* < 0.05, respectively). Additionally, they investigated the effect of silencing *ADAM10* in the presence and absence of DHT and showed that siRNA targeting *ADAM10* significantly reduced the proliferation of LNCaP cells at day 4 (*p* < 0.01) in the absence of DHT. However, a massive decrease in cell growth was observed on day 3 (*p* < 0.01) and day 4 (*p* < 0.001) in the presence of DHT.

## 4. Discussion

This is the first systematic review to investigate the literature on the impact of androgens on APP and its cleavage enzymes in relation to cancer. Five studies were included in the review. Although some of these studies were published over 20 years ago, they were the first to explore how androgens might influence APP and α-secretase enzymes such as ADAM9, ADAM10, and ADAM17, that collectively give rise to a similar cleavage product of APP called sAPPα. Three studies showed that DHT increased *ADAM10* expression in prostate cancer cells and stimulated its nuclear translocation. The remaining two studies reported that androgens upregulated *APP* expression in both prostate and breast cancer cells (Figure 4). Despite a rigorous search of scientific databases, the most recent study we found was published in 2013, which highlights the lack of recent research in this area, despite these promising findings early on. There are likely many reasons why little has been published regarding the impact of androgen on APP processing pathways in cancer. APP has been extensively studied in the context of AD and is well known in the field of neurodegeneration; however, despite being widely expressed peripherally, limited research has been carried out to ascertain APP’s function outside of the brain. Whilst the inverse association between AD and cancer has been reported, the mechanisms underpinning this have not yet been established. There will be many important pathways that contribute to this inverse association. Therefore, whilst the potential for understanding the regulation of APP by androgens in cancer could be very important, it is likely that more classically cancer-specific pathways have been prioritised by the cancer research community over APP, which has predominantly been previously thought to only play a role within the central nervous system. As our review highlights, since the early reports on APP in relation to androgens, there has been follow-up, but it is limited, and hopefully, with better collaborative approaches between researchers in AD and cancer, this is one pathway that warrants further investigation in the cancer space.

Although we looked for all types of studies, we found a lack of evidence on downstream signalling pathways, only finding gene expression-level data. Additionally, nearly all the studies we found were investigating prostate cancer, with one study investigating breast cancer, showing evidence gaps in this area for all other cancer types. There is also a high risk for publication bias. As we only searched for published literature, there is a possibility that there have been relevant, more recent studies in this field that have not been published due to neutral findings, that we are not aware of.

Androgens (testosterone and DHT) are essential for the normal development and functioning of the prostate [31] and also play a role in prostate cancer initiation and progression [32]. Serum DHT levels in men between 18–59 years normally range from 0.47 to 2.65 nM/L, while in older individuals 71–87 years, levels range from 0.49 to 3.2 nM/L [33]. Therefore, concentrations of DHT and AR agonists described in the included studies reflect both normal and pathophysiological levels, to correlate phenotypic effects, such as increased cell growth and molecular changes in APP processing, with disease characteristics.

The two studies by McCulloch et al. assessed the impact of DHT on cell growth and demonstrated the anticipated increase in cell proliferation in response to DHT in androgen-responsive prostate cancer cells, LNCaP [26,27]. At comparable doses of an androgen analogue, R1881, in the same cell line, Takayama et al. later reported an increase in mRNA levels of *APP* [29]. This increase in *APP* mRNA in response to DHT was also observed in breast cancer cells [30]. A positive association was also observed between APP and AR in breast cancer tissues. These reports suggest that the growth effects of androgens were associated with increases in APP in cancer cells. However, future investigations are needed to identify whether androgen-induced APP processing is toward the non-amyloidogenic or amyloidogenic pathway.

The study by Takayama et al. provided the link between androgens and APP through identifying an AR binding site in the 3′ downstream region and intron 1 of APP using the CHIP-chip method, indicating APP as an androgen-targeted gene [29]. However, the study by Takagi et al. did not confirm the nature of the AR–APP interaction [30]. Therefore, it remains unclear whether APP is primarily regulated directly by AR or whether secondary androgen-responsive pathways also contribute to its regulation.

To understand the role of APP itself in proliferation, gain-of-function and loss-of-function studies were performed. Overexpressing APP resulted in a marked increase in the proliferation of LNCaP cells. In contrast, knocking down APP inhibited tumour growth in vivo and in vitro. In breast cancer cells, silencing APP reduced cell growth and it was additionally noted that there was no induction of apoptosis [30]. These findings highlight that there might be a crucial role of APP in cancer growth. APP is a very complicated protein as for example, it has different isoforms, cleavage products and interacting proteins. The status of APP, and therefore its contribution to cellular proliferation, will be different depending on the cellular context. Further supporting the relevance of APP in proliferation, Takayama et al. found that treating with recombinant soluble APP, which can be naturally generated as the secreted NH2-terminal ectodomain of APP through cleavage at the α-secretase site, significantly increased cellular proliferation [29]. This emphasized that APP exerts its growth-promoting effects via the non-amyloidogenic pathway. Another study suggests that the cleavage peptide, sAPPα, could be the main factor that promotes breast cancer progression mediated by APP [20]. Given these findings, understanding how androgen signalling influences APP processing is crucial, as it may reveal novel therapeutic targets.

Whilst the effect of androgens on APP is the focus of this review, it is helpful to investigate the effect of their inhibitors to explore for corroborating effects. These drugs offer a level of helpful precision in answering questions about the androgens since they have a higher affinity to attach to the AR than androgens (testosterone and DHT) and they additionally inhibit nuclear translocation of the complex to block gene transcription [34]. Takagi et al. reported that an AR antagonist, hydroxyflutamide, decreased *APP* mRNA levels in MCF7 cells [24,30]. It is classified as a first-generation anti-androgen [35,36]. However, acquired resistance develops, leading to disease progression into castration-resistant prostate cancer [36]. Second-generation anti-androgens were developed to evade such resistance including enzalutamide and apalutamide [34]. Thus, they have been used on patients with nonmetastatic and castration-resistant prostate cancer. The PROSPER trial assessed enzalutamide and the SPARTAN trial evaluated apalutamide: both significantly improved metastatic-free survival and delayed the risk of developing an aggressive phenotype [37,38]. Despite the interesting initial studies, the effect of current anti-androgens on APP and cleavage enzymes remains largely unexplored and offers a potentially important research opportunity.

In various experimental paradigms, DHT has been shown to regulate the homeostasis between cellular growth and programmed cell death (apoptosis) via promoting additional growth factor production (GF), including EGF, IGFs, and keratinocyte growth factor (KGF), that activate important signalling pathways, for instance, PI3K/AKT and Ras/MAPK pathways [39,40]. Tang et al. solely reported that tumour growth factor-α (TGF-α) upregulated levels of APP in nasopharyngeal carcinoma cells due to EGFR activation [41]. Currently, no studies have investigated the effect of IGF-I on *APP* expression in cancer. However, IGF-I has been shown to induce APP production in AD [42,43]. Importantly, such growth factors play important roles in the growth and survival of cancer cells at all stages of progression and are strongly associated with aggressive prostate cancer [40,44], suggesting that the relevance of APP may not be limited to androgen-responsive tumours. The link between the growth-promoting effects of androgens and the associated increase in APP provides impetus to understand the androgenic regulation of APP processing pathways and its potential implications for cancer. In the study by McCulloch et al., 2004 the influence of DHT, IGF-I, and EGF on *ADAM10* expression and synthesis was assessed [27]. The combination of DHT and IGF-I increased *ADAM10* mRNA levels by 1.8-fold, while EGF alone raised them by 1.7-fold. IGF-I plus DHT also upregulated the immature form of ADAM10 protein level by 1.8-fold and the mature form by 3–4-fold. EGF increased the protein abundance of immature ADAM10 by 3-fold and mature ADAM10 by 4-fold. This study highlighted a more comprehensive influence of three key growth promoters, DHT, IGF-I, and EGF, on *ADAM10* expression and abundance, which cleaves APP as part of the non-amyloidogenic pathway, in an AR-dependent cell line.

Three studies emphasized the impact of DHT on increasing *ADAM10* expression and abundance. However, the role of ADAM10 as an APP-cleaving enzyme in response to DHT remains underexplored in cancer. ADAM10 is known for its shedding activity, particularly in the release of EGFR ligands, and is upregulated in response to EGF and IGF-I, both of which are implicated in aggressive prostate cancer [40,45]. Notably, the addition of IGF-I to DHT significantly increased ADAM10 levels, suggesting a regulatory link between androgens, growth factors, and APP processing.

Given that ADAM10 is the primary α-secretase responsible for non-amyloidogenic APP cleavage, understanding how androgen signalling influences this pathway could reveal novel mechanisms of tumour progression [16]. Moreover, ADAM10′s nuclear translocation and interaction with AR raise questions about its potential transcriptional role in androgen signalling. Investigating whether androgen-induced ADAM10 upregulation shifts APP processing toward sAPPα production and how this affects cancer biology will be critical for determining whether targeting and enhancing ADAM10 could serve as a therapeutic strategy for cancer. In addition, understanding the mechanisms of sAPPα that contribute to cancer proliferation could potentially lead to targeting sAPPα as a new therapeutic target.

## 5. Conclusions

In conclusion, this review highlights the potential role of androgen regulation in APP and its processing in cancer, suggesting that APP and α-secretase enzymes may be modulated by androgens (Figure 4) and could contribute to disease progression in breast and prostate cancers. However, recent evidence is lacking and requires more research: Future investigations should clarify how androgens influence APP processing through either amyloidogenic or non-amyloidogenic pathways by assessing changes in the enzymes involved. It will also be important to explore the effects of androgen receptor inhibitors used in cancer therapy on APP pathways, and to determine whether the products of these pathways, including sAPPα or sAPPβ, contribute to disease development. Moreover, extending this research to other malignancies, such as bladder, lung, pancreatic, and colorectal cancers, will be essential to establish whether APP processing represents a broader mechanism in cancer biology and to identify new opportunities for therapeutic intervention.

## Figures and Tables

**Figure 1 cimb-47-01041-f001:**
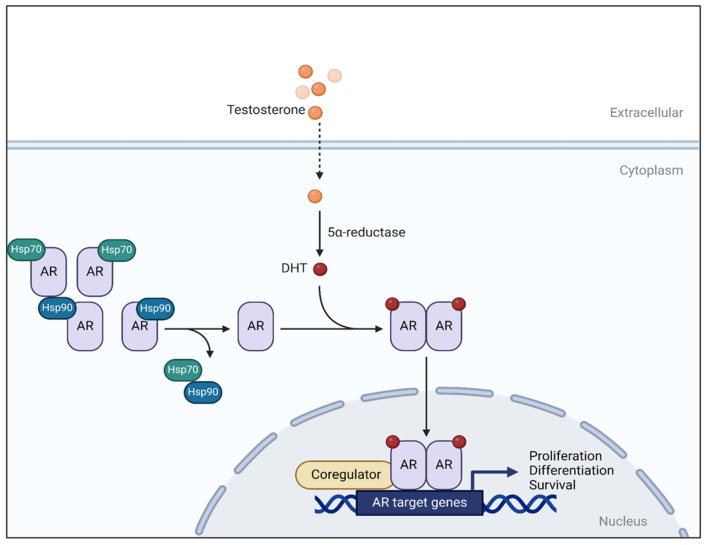
Androgen receptor activation pathway. Testosterone (primary androgen) is converted into 5α-dihydrotestosterone (DHT) *via* 5α-reductase enzyme. DHT attaches to the AR with high affinity, leading to the dissociation of heat shock proteins from the AR. The ligand-receptor complex translocates into the nucleus. In the nucleus, AR dimerizes and binds to androgen response elements (ARE) in the promoter region of the target gene that promotes the growth, development, and survival of prostate cells. Adapted from [13] with modifications and created with BioRender.com.

**Figure 2 cimb-47-01041-f002:**
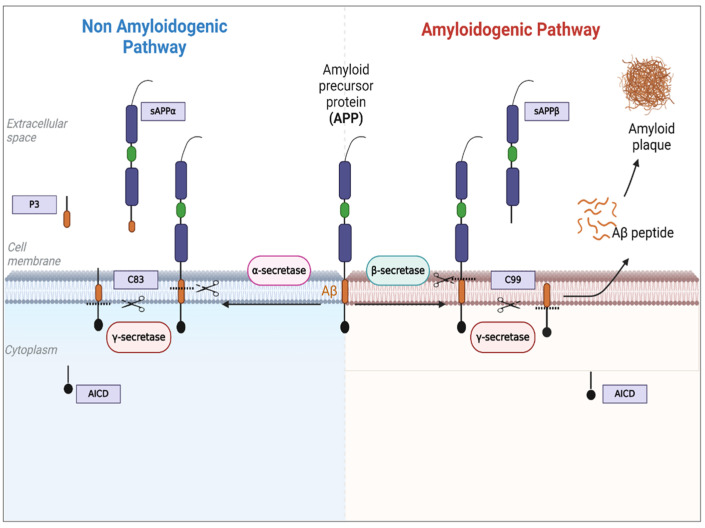
Amyloid precursor protein (APP) processing pathways. There are two types of APP proteolysis pathways: non-amyloidogenic and amyloidogenic. The non-amyloidogenic begins with α-secretase cleavage to release soluble amyloid precursor (sAPPα) and C-terminal 83, which is then cleaved by γ-secretase to produce P3 and APP intracellular domain (AICD). α-secretases involved in this process include: the disintegrin and metalloprotease family (ADAMs), which includes ADAM9, ADAM10, ADAM17 or tumour necrosis factor-alpha converting enzyme (TACE), and the aspartyl protease, beta-site APP cleavage enzyme 2 (BACE2). The amyloidogenic route begins with N-terminal cleavage by β-secretase or BACE1 (β-secretase APP converting enzyme 1) to yield soluble amyloid precursor protein (sAPPβ) and C-terminal 99. This is followed by γ-secretase cleavage, which results in the production of Aβ peptides and AICD. The neurotoxic Aβ peptide creates insoluble fibrils that form amyloid plaques in the brain, resulting in AD. Adapted from [14] with modification and created with BioRender.com.

**Figure 3 cimb-47-01041-f003:**
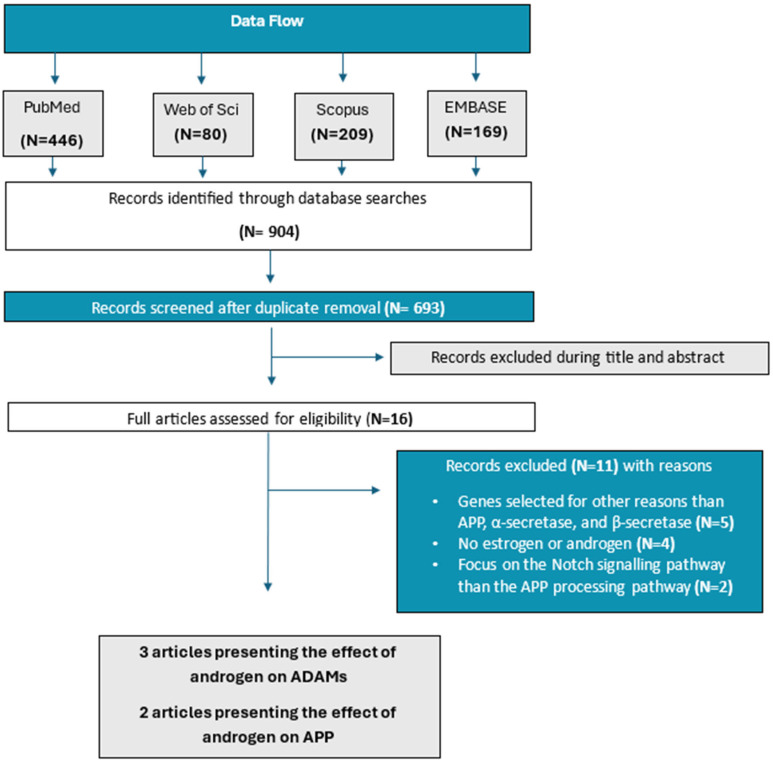
Flowchart of the screening process for study inclusion and exclusion.

**Figure 4 cimb-47-01041-f004:**
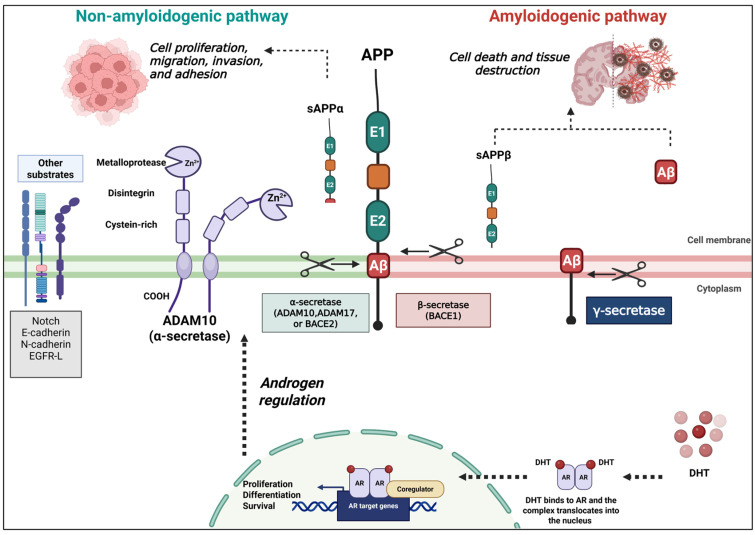
Schematic representation of APP processing pathways and ADAM10 function. APP can undergo non-amyloidogenic processing by α-secretases (ADAM10, ADAM17, or BACE2), resulting in the generation of sAPPα, which has been linked to cell proliferation, migration, survival, and neuroprotection. Alternatively, amyloidogenic processing by β-secretase (BACE1) and the γ-scretase comlex produces sAPPβ and amyloid-β (Aβ) peptides, implicated in neurodegeneration; however, their role in cancer remains limited and conflicting. ADAM10, a zinc-dependent metalloprotease, also cleaves additional substrates such as Notch, N-cadherin, and E-cadherin, thereby influencing adhesion and signaling in cancer biology. The androgen receptor (AR) may regulate APP expression and α-secretase activity, suggesting a potential link between androgen signaling, APP processing, and cancer progression. Green = non-amylodiogenic pathway; Red = amyloidogenic pathway; Dashed arrow = regulatory influence. Created with BioRender.com.

**Table 1 cimb-47-01041-t001:** Characteristics of the selected studies.

Author (Year)	Study Title	Cancer Type	Experimental Techniques	Study Outcomes
(McCulloch et al., 2000) [26]	The expression of the Adams proteases in prostate cancer cell lines and their regulation by dihydrotestosterone	Prostate	Prostate cancer (PCa) cell line, reverse transcription polymerase chain reaction (RT-PCR), northern blotting, and MTT (3-[4,5-dimethylthiazol-2-yl]-2,5-diphenyl tetrazolium bromide) assay	DHT (0.1 to 10 nM/L) increased mRNA levels of ADAM 10. DHT reduced the mRNA level of ADAM17 in LNCaP cells.
(McCulloch et al., 2004) [27]	Expression of the Disintegrin Metalloprotease, ADAM-10, in Prostate Cancer and Its Regulation by Dihydrotestosterone, Insulin-Like Growth Factor I, and Epidermal Growth Factor in the Prostate Cancer Cell Model LNCaP	Prostate	(PCa) cell line, immunohistochemistry (IHC), MTT assay, western blotting (WB), RT-PCR, and assessment of molecules in nuclear and cytoplasmic protein fractions	DHT (10 nM/L) and insulin-like growth factor-I (IGF-I) (50 ng/mL) combined significantly increased ADAM10 mRNA and protein levels. Epidermal Growth Factor (EGF) (50 ng/mL) increased ADAM10 mRNA levels.
(Arima et al., 2007) [28]	Nuclear translocation of ADAM-10 contributes to the pathogenesis and progression of human prostate cancer.	Prostate	(PCa) cell line, IHC, WB, nuclear and cytoplasmic protein fractions, immunoprecipitation (IP), immunocytochemistry (ICC), RT-PCR, small interfering RNA (siRNA) transfection, and MTT assay.	DHT (0.1 to 100 nM/L) increased the nuclear protein level of ADAM10 in LNCaP cells. DHT enhanced the translocation of ADAM-10 from the cell membrane and cytoplasm to the nucleus.
(Takayama et al., 2009) [29]	Amyloid Precursor Protein Is a Primary Androgen Target Gene That Promotes Prostate Cancer Growth	Prostate	(PCa) cell line, WB, RT-PCR, RNA Sequencing, siRNA transfection, proliferation assay, luciferase assay, chromatin immunoprecipitation (ChIP DNA), IHC, and ICC	The androgen methyltrienolone (R1881), an AR agonist (10 nM/L), upregulated APP mRNA levels and protein abundance in LNCaP cells. APP overexpression enhanced LNCaP cell proliferation. Silencing APP reduced cell growth
(Takagi et al., 2013) [30]	Amyloid precursor protein in human breast cancer: An androgen-induced gene associated with cell proliferation	Breast	Breast cancer cell lines, IHC, RT-PCR, WB, scoring immunoreactivity, siRNA transfection, MTT assay, and apoptosis assay (Tali^TM^ apoptosis kit)	DHT (10 nM/L) increased the expression of APP mRNA in MCF7 cells. An AR blocker, hydroxyflutamide, suppressed the mRNA levels of APP.

**Table 2 cimb-47-01041-t002:** Criteria applied for assessing the quality of included studies (N/A = not applicable).

Rating Criteria for Study Quality		McCulloch et al., 2000 [26]	McCulloch et al., 2004 [27]	Arima et al., 2007 [28]	Takayama et al., 2009 [29]	Takagi et al., 2013 [30]
Use of In Vivo Models	**5:** The study includes robust in vivo experiments using relevant animal models. **3:** The study uses in vivo models, but the relevance of the model to the research question is limited. **N/A:** No in vivo experiments are included.	**N/A**	**N/A**	**N/A**	**3**	**N/A**
Use of Human Tissue Samples	**5:** The study analyses fresh or archival human tissue samples with direct relevance to the research question. **3:** Tissue samples are used but lack relevance (e.g., not disease-specific). **N/A:** No tissue samples are used.	**N/A**	**5**	**5**	**5**	**5**
Analysis of Clinical Data	**5:** The study integrates clinical data, including patient demographics, outcomes, or treatment effects. **3:** Clinical data are included but lack statistical depth or relevance to the research question. **N/A:** No clinical data are analysed.	**N/A**	**N/A**	**5**	**5**	**5**
Measurement of mRNA and Protein Expression	**5:** Both mRNA and protein levels are measured using robust techniques **3:** Only mRNA or protein is measured, not both. **N/A:** No expression/abundance data are included.	**3**	**5**	**5**	**5**	**5**
Methodological Complexity	**5:** The study employs a diverse range of advanced and complementary methodologies (e.g., IF, CHIP-chip, IP, ICC, IHC, MTT, or knockdown). **3:** A moderate number of methods are used, but their application is somewhat limited, affecting the depth of analysis. **N/A:** The study relies on a single or a few basic methodologies, with minimal experimental complexity.	**3**	**5**	**5**	**5**	**5**
Mechanistic Insights	**5:** The study identifies and explains mechanisms mediating the observed effects, supported by experimental evidence (e.g., pathway analysis, inhibitor studies). **3:** Mechanisms are hypothesised but not directly tested. **N/A:** No mechanistic explanations are provided.	**N/A**	**N/A**	**3**	**3**	**3**
**Total**	**23–30:** High quality. **20–22:** Moderate quality. **Below 20:** Low quality.	**6**	**15**	**23**	**26**	**23**

## Data Availability

No new data were created or analyzed in this study.

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
