# Peer review of "Androgen Effects on Amyloid Precursor Protein Processing Pathways in Cancer: A Systematic Review"

_cimb, 2025, doi:10.3390/cimb47121041_

Round 1

Reviewer 1 Report

Comments and Suggestions for Authors

In the review manuscript titled "Androgen effects on Amyloid Precursor Protein Processing 2 Pathways in Cancer: A Systematic Review” by Alhadrami et al,  the authors conducted a systematic review of studies examining how androgens influence APP and its processing enzymes in cancer. They screened literature from 2000–2024 and identified five relevant studies in prostate and breast cancer models. Their analysis showed that androgens upregulate APP and ADAM10, suggesting a role in promoting tumor progression. While the review is interesting and discusses important pathways in cancers, there are several points to consider critically before the manuscript could be considered for publication.

1) There are some spelling mistakes and typing errors that authors need to correct
2)  The review does not address the potential for publication bias. While a funnel plot was not feasible due to the small number of studies and lack of meta-analysis, a narrative discussion of this risk is crucial.
3)  The quality and risk-of-bias assessment was conducted using a custom-developed set of criteria. The use of a non-validated tool, however, may affect the standardization of the process. To enhance methodological rigor, the authors could employ established tools such as SYRCLE and ROBINS-I.
4) The review presents APP as androgen-regulated but does not critically assess if this is a direct transcriptional event or an indirect effect via secondary AR-activated pathways. This is crucial for understanding the primary mechanism.
5) The authors need to explain whether APP’s proliferative effects are cell-line specific.  
6)  The authors provide evidence that relies heavily on older studies than newer, which loses their relevance given the advancements in modern AR biology and cancer genomics.  

Author Response

Thank you for the thorough and constructive comments provided on our manuscript entitled “Androgen effects on Amyloid Precursor Protein Processing Pathways in Cancer: A Systematic Review” (Manuscript ID: cimb-3987548). We have carefully revised the manuscript in accordance with the comments and suggestions provided. Below, we provide a detailed, point-by-point response. Reviewer comments are presented in bold, followed by our responses.

Comment 1:

There are some spelling mistakes and typing errors that the authors need to correct.

Response: We thank the reviewer for highlighting this. We have thoroughly proofread the manuscript and corrected spelling, typographical, and formatting errors throughout the text.

Comment 2:

The review does not address the potential for publication bias. While a funnel plot was not feasible, a narrative discussion is crucial.

Response: We have added a narrative discussion to the end of the first paragraph in the Discussion: “There is also a high risk for publication bias. As we only searched for published literature, there is a possibility that there have been relevant, more recent studies in this field that have not been published due to neutral findings, that we are not aware of.”

Comment 3:

The quality and risk-of-bias assessment was conducted using a custom-developed set of criteria. The use of a non-validated tool may affect standardization. Consider SYRCLE or ROBINS-I.

Response: We appreciate this comment. After evaluating these tools, we concluded that SYRCLE (animal studies) and ROBINS-I (non-randomized human studies) were not applicable to our dataset, which consists mainly of in vitro mechanistic studies and tissue-level analyses. As no validated risk-of-bias tool exists for such study designs, we created a structured checklist tailored to experimental rigor in cell-based cancer studies. We have clarified this rationale in the Methods (Risk of Bias) section.
(Clarification added in Methods 2.4.)

Comment 4: The review presents APP as androgen-regulated but does not critically assess whether this is a direct or indirect mechanism.

Response: We thank the reviewer for this important point. We have now added a clarification to the discussion section (lines 325-330).

 Comment 5:

The authors need to explain whether APP’s proliferative effects are cell-line specific.

Response: APP is a very complicated protein as for example, it has different isoforms, cleavage products and interacting proteins. The status of APP and therefore its contribution to cellular proliferation, will therefore be different depending on the cellular context, which will be cell-line specific. We have added a sentence to highlight this in the discussion (lines 336-339).

 Comment 6:

The authors rely heavily on older studies, which may limit relevance given modern AR biology.

Response: We can only synthesise evidence that is available in the published literature and was found by our rigorous literature search methods. The fact that recent evidence in this area is limited is a finding that is important for us to highlight as needing more research. We have clarified this point in the Conclusion: “However, recent evidence is lacking and requires more research:……..”

Reviewer 2 Report

Comments and Suggestions for Authors

This is a well executed and rather readable, but also very systematic review article and it provides information that is partly limited...by the lack of data! And that needs to be considered here carefully.

It doesnt cover an "ancient" area of research thats covered by hundreds of studies. There is literally just a handful of studies - which makes you also wonder: is this field ready to be reviewed ? Whats there to report, if there are so little studies? IN addition - the studies that exist are all dating to 2013 or earlier... nothing has happened in over 10 years. This significant gap is - by itelsf - actually a finding on its own, and not any kind of oversight, sloppiness, or even a methodological flaw. The authors should even emphasize this a bit more because I consider this significant.

And instead also discuss a bit deeper: Why is this so? Has the research field lost interest? Is the association and functional connection debunked, for any reason? 

So the review is actually largely occupied with reporting knowledge GAPs - not exactly a flood of findings. Instead, the authors find room to describe how they have systematically looked into the quality of the 5 or 6 studies they are reporting: this is shown in Table 2. For me, this is unique; maybe it would hel also in larger studies that report about more thn 5-6 papers; but would it be shown then? This is something that I would expect in supplemental data. 

Since the scope of the review is naturally very narrow, I wonder if the authors should maybe consider slightly expanding the topic - beyond the 3 core "agents" APP, AR, and ADAM10. That would give them more substance without major jumps in different territories. For example, report on the ADAM10/AR axis in other cancer types like melanoma, for which there are newer data. Maybe also look into beta-secretases, too? ANd maybe look if APP does something in cancers on its own, regardless of the AR/androgens connection? 

There are a few formatting and style issues: small things like "397.00 deaths" (should be 390.000), "heter-ogeneity"  should be corrected to "heterogeneity". Theres a few more of these but its not a big issue. 

Comments on the Quality of English Language

There are a few formatting and style issues: small things like "397.00 deaths" (should be 390.000), "heter-ogeneity"  should be corrected to "heterogeneity". Theres a few more of these but its not a big issue. 

Author Response

Thank you for the thorough and constructive comments provided on our manuscript entitled “Androgen effects on Amyloid Precursor Protein Processing Pathways in Cancer: A Systematic Review” (Manuscript ID: cimb-3987548). We have carefully revised the manuscript in accordance with the comments and suggestions provided. Below, we provide a detailed, point-by-point response. Reviewer comments are presented in bold, followed by our responses.

Comment 1:

There are only a handful of studies—Is the field ready to be reviewed? This very small number of studies is itself a finding.

Response: We agree that the very small number of studies is itself a finding and have further clarified this in our Conclusions. It is only by systematically reviewing the literature (as we have done) that we can highlight this important finding and inform future researchers of where the gaps are that need filling.

 Comment 2:

Why has nothing been published in 10 years? Has the field lost interest? Was the mechanism debunked? This should be discussed.

Response: This is an excellent point that the reviewer has raised and one which deserves more attention in the discussion.  We have added an additional paragraph (Lines 285-299), ‘There are likely many reasons why little has been published regarding the impact of androgen on amyloid precursor protein (APP) processing pathways in cancer. APP has been extensively studied in the context of Alzheimer’s disease (AD) and is well known in the field of neurodegeneration, however, despite being widely expressed peripherally, limited research has been carried out to ascertain APP’s function outside of the brain. Whilst the inverse association between AD and cancer has been reported, the mechanisms underpinning this have not yet been established. There will be many important pathways that contribute to this inverse association. Therefore, whilst the potential for understanding the regulation of APP by androgens in cancer could be very important, it is likely that more classically cancer-specific pathways have been prioritised by the cancer research community over APP. which has predominantly been previously thought to only play a role within the central nervous system. As our review highlights, since the very early reports on APP in relation to androgens, there has been follow-up, but it is limited and hopefully with better collaborative approaches between researchers in AD and cancer, this is one pathway that warrants further investigation in the cancer space’.

Comment 3:

Table 2 might be more appropriate as supplemental material.

Response: We appreciate the suggestion. We chose to keep Table 2 in the main text because the entire review is built around the quality assessment of a very small number of studies; however, we are happy to move it if the editor prefers.

 Comment 4:

The scope is narrow; consider slightly expanding the topic beyond APP/AR/ADAM10, such as including melanoma, β-secretases, or APP-related cancer functions independent of androgens.

Response: We thank the reviewer for this thoughtful suggestion. However, the scope of our review was intentionally predefined based on our PICO question, which aimed to determine whether androgen interventions influence APP processing pathways and subsequently impact cancer proliferation and development. The search did include beta-secretases as suggested by the reviewer.  Because our research question specifically focuses on androgen-driven modulation of APP and its cleavage enzymes, we cannot expand the topic to include APP-related cancer functions that occur independently of androgen signaling, as this would fall outside the aims and structure of our systematic analysis.

Regarding the recently published melanoma paper examining ADAM10 and AR, this study appeared in January 2025, which is outside the predefined search window of our review (01/01/2000–31/12/2024). Including this article would require restarting the entire screening and selection process to maintain methodological integrity. For this reason, it cannot be incorporated into the current systematic review.

Finally, our search strategy already included all major enzymes involved in APP processing, as reflected in our comprehensive search terms across four databases (PubMed, Scopus, Web of Science, EMBASE). This included androgen effects on BACE1, BACE2, ADAM10, ADAM17, TACE, and other metalloproteases, ensuring that all potential androgen–APP processing interactions were captured within the limits of our defined research question.

Comment 5:

Formatting issues such as number formatting (“397.00 deaths”) should be corrected.

Response: We have corrected these formatting inconsistencies throughout the manuscript, and the word “heterogeneity” was already written in its correct format. No correction was required for this term.

Round 2

Reviewer 1 Report

Comments and Suggestions for Authors

The authors have addressed all my concerns and have made suggested changes in the manuscript. I, therefore, recommend the manuscript suitable for publication in its revised form.